# A universal method for the purification of C2H2 zinc finger arrays

**Jingchang Liang**[1,2,3], **Maia Azubel**[1], **Guanqiao Wang**[3], **Yan Nie**[3], **Roger D. Kornberg**[1], **Andrew J. Beel**[1]*, **Pierre-Jean Mattei**[1]*

**1** Department of Structural Biology, Stanford University, Stanford, California, United States of America, **2** WLA Laboratories, Shanghai, China, **3** Shanghai Institute for Advanced Immunochemical Studies, ShanghaiTech University, Shanghai, China

* pmattei@stanford.edu (P-JM); beelaj@stanford.edu (AJB)

**Data Availability Statement:** All relevant data are within the manuscript and its Supporting Information files.

**Funding:** AJB was funded by an NIH grant 1DP5OD033431 J.L was funded by WLA

## Abstract

Zinc fingers (ZFs) are compact, modular, sequence-specific polynucleotide-binding domains uniquely suited for use as DNA probes and for the targeted delivery of effector domains for purposes such as gene regulation and editing. Despite recent advances in both the design and application of ZF-containing proteins, there is still a lack of a general method for their expression and purification. Here we describe a simple method, involving two chromatographic steps, for the production of homogeneous, functional ZF proteins in high yield (one milligram per liter of bacterial culture), and we demonstrate the generality of this method by applying it to a diverse set of eight C2H2-type ZF proteins. By incorporating a surface-exposed terminal cysteine residue that enables site-specific conjugation with maleimide-activated fluorophores, we confirm the suitability of these probes for *in situ* labeling of specific DNA sequences in human cells.

## Introduction

The zinc finger (ZF), discovered by Klug and colleagues as a zinc-coordinating, polynucleotide recognition element in transcription factor IIIA of *Xenopus laevis* [1], is the basis for the largest family of transcription factors in the genomes of vertebrates [2]. They are present in 2–3% of all human protein-coding genes [3], including nearly half of all annotated transcription factors [2]. ZFs exhibit a wide variety of structural motifs, denominated by the number of cysteine and histidine residues coordinating the central zinc ion, among which the most common and widely studied is the C2H2 type containing two cysteines and two histidines [4, 5]. C2H2 ZFs comprise 28–30 amino acids assuming a zinc-stabilized ββα fold, consisting of two short β-strands followed by one α-helix. The latter "recognition helix" confers sequence specificity by forming hydrogen bonds, in the major groove of DNA, to three consecutive base pairs on one strand and a preceding base pair on the opposite strand [6, 7].

Importantly, ZFs function as modular recognition elements, and can be assembled in tandem to form polydactyl arrays capable of recognizing extended sequences [5, 8–13]. Indeed, among naturally occurring human C2H2 ZF arrays, the number of fingers varies from three to about three dozen [14]. Extensive work has been done to understand the determinants of

laboratories. Y.N. and G.W. were funded by a grant from ShanghaiTech University. The funders had no role in study design, data collection and analysis, decision to publish, or preparation of the manuscript.

**Competing interests:** The authors have declared that no competing interests exist.

sequence specificity (or "rules") of individual ZFs [15]. Exploiting knowledge of such rules to design ZF modules for the recognition of extended sequences, however, is not straightforward, because the modules are not independent: the specificity of each ZF is influenced by adjacent ZFs within an array, due to overlap of the base-contacting residues [16]. Nevertheless, recent advances in deep learning have enabled the development of artificial-neural-network models capable of designing polydactyl arrays for arbitrary DNA sequences [17]. This technology is readily applicable in cases where ZF proteins can be endogenously expressed by the cells of interest, for example for gene regulation by fusion of ZF arrays to VP64 or KRAB domains [18, 19], for gene editing by fusion to domains of the FokI nuclease [20, 21], or for DNA localization by fusion to GFP or other tags [22, 23]. In contrast, the application of this technology is presently limited in cases requiring exogenous ZF proteins due to the lack of a general procedure for their expression and purification. Exogenous ZF nucleases, for instance, offer comparatively greater gene-editing efficiency with lower off-target effects [24].

Many protocols for ZF protein expression and purification have been reported [6, 24–36]; however, these protocols are almost invariably tailored to specific ZF proteins and must be modified to achieve satisfactory results with other ZF proteins. These protocols vary in many respects, including the selection and placement of an affinity or solubilization tag, the conditions of expression, the chromatographic steps, and so forth; consequently, there is no clear basis upon which to develop a general method. Affinity or solubilization tags reported for purification of ZF proteins include polyhistidine (His tag) [25, 27, 28], glutathione S-transferase (GST) [29, 30], and maltose-binding protein (MBP) [31, 32]. Others have forgone a tag by exploiting intrinsic properties of ZF proteins such as their affinity for $Zn^{2+}$ [33] or the cognate DNA sequence [34]. In some cases, the expressed protein is soluble [25, 27, 28], while in other cases it must be recovered from the insoluble fraction [35, 36].

A general method for the purification of ZF proteins would be especially beneficial for the development of probes for non-repetitive DNA sequences. A single species of ZF protein often suffices to detect repetitive sequences, as the repeats locally concentrate probes, providing a convenient mechanism for signal amplification. In contrast, for the detection of nonrepetitive sequences, a single species of ZF proteins is generally insufficient, and one may need to rely on the simultaneous binding of a set of adjacent, perhaps tiling, ZF proteins. The need to determine suitable purification conditions for each new ZF protein is a major impediment to the development of probes for non-repetitive sequences.

Here we describe a general method for the expression and purification of milligram quantities of functional C2H2 ZF arrays (polypeptides composed of one or more ZF motifs). To demonstrate its generality, we applied the method to eight different ZF arrays, purifying each to homogeneity. Following purification, we conjugated a maleimide-activated fluorophore to a terminal cysteine of each protein. Finally, we demonstrate the detection of telomeric DNA in fixed and permeabilized human cells, confirming the utility of these probes for *in situ* detection and localization of DNA sequences.

## Results

To support the development of a general method for the expression and purification of C2H2-family ZF arrays, we compiled a diverse set of previously characterized constructs of both natural and synthetic origin (Table 1 and S1 Table). The selected ZF arrays varied across multiple dimensions, including number of fingers (1 to 6), molecular weight (6.6 to 22.5 kDa), isoelectric point (7.8 to 10.6), and hydrophobicity (−1.5 to −0.5 on the Kyte-Doolittle scale [37]) (Table 1). We began by testing a straightforward design—previously shown to be effective for, among other examples, a ZF nuclease targeting the beta-lactoglobulin locus [25], the

**Table 1. Zinc finger arrays used in this study.**

| DNA locus | ZF | Target DNA Sequence (5'-3') | Molecular Weight (kDa) | # of ZFs | pI | GRAVY | Yield (mg/L) |
|---|---|---|---|---|---|---|---|
| *ccr5* | CCR5L [24, 38] | GTCATCCTCATC | 14.76 | 4 | 10.20 | -0.586 | 0.9 |
| | CCR5R [24, 38] | AAACTGCAAAAG | 14.86 | | 9.81 | -0.583 | 2.1 |
| *cxcr4* | CXCR4L [24, 39] | GTAGAAGCGGTC | 14.17 | | 10.39 | -0.699 | 1.6 |
| | CXCR4R [24, 39] | GACTTGTGGGTG | 14.28 | | 10.20 | -0.687 | 1.7 |
| *vegfa* | ZVEGF [40] | GGGGGTGAC | 10.56 | 3 | 9.50 | -1.478 | 1.4 |
| telomere | TZAP$_{11}$ [41] | TTAGGG | 6.60 | 1 | 9.57 | -1.128 | 1.2 |
| | TZAP$_{9-11}$ [41] | | 19.60 | 3 | 7.76 | -0.509 | 1.5 |
| *brf1* | ZBrf1 [17] | CGCCCAGCTGGGGGCGGGGGA | 22.47 | 6 | 10.56 | -0.862 | 1.0 |

DNA-binding domain of AreA [27], and ZNF191 [28]—which consisted of an N-terminal His tag followed by the coding sequence of a ZF array from S1 Table. However, the expression was low (Fig 1A), as was the yield (less than 0.1 mg of protein per liter of culture).

Seeking to increase the yield, we explored the possibility of a larger tag, and initially opted for GST, which has also been used for the expression and purification of ZF proteins [29, 30]. Although expression levels of GST fusions were 30 times greater than those of their His-tag-only counterpart (Fig 1A), they were not consistently soluble, and often the majority of the heterologous protein was present in the pellet after cell lysis (Fig 1B).

Seeking to develop a method that avoids a requirement for refolding, we tested another tag that is commonly used to improve protein solubility and expression, the small ubiquitin-like modifier protein (SUMO) from *S. cerevisiae*, Smt3 [42]. For affinity purification, a His tag was fused to the N terminus of SUMO. IPTG induction led to a prominent band of the expected molecular weight in whole cell lysates (Fig 1A, compare lanes with and without IPTG). The addition of a SUMO tag resulted in a 50-fold increase in the expression of heterologous protein (Fig 1A), and Ni-NTA resin retained 45-fold more heterologous protein from the soluble fraction of cell lysates containing His-SUMO-tagged protein than those containing His-tagged protein (Fig 1C).

We devised a purification strategy consisting of two steps: (i) capture on Ni-NTA resin, coupled with on-column cleavage by Ulp1, and (ii) cation exchange on SP Sepharose resin (Fig 1D). Following this procedure for CXCR4L, the fusion protein was immobilized on Ni-NTA (Fig 1E, lane R), cleaved efficiently, and recovered in high yield (Fig 1E, lane E). Further purification by cation-exchange chromatography afforded a homogeneous product (Fig 1F), with an overall yield of 1.6 mg of CXCR4L per liter of culture. The addition of L-arginine during cleavage is crucial: it increases the cleavage efficiency by two-fold (compare the band for SUMO-ZVEVGF in lanes "BC" and "AC" in the presence [+] and absence [−] of L-arginine in Fig 2) and decreases the retention of cleaved protein by over three-fold (compare the band for ZVEGF in lanes "AC" and "E" in the presence [+] and absence [−] of L-arginine in Fig 2), enhancing the recovery of cleaved protein by over six-fold. Furthermore, by including L-arginine throughout the purification protocol, the yield of purified protein from one liter of culture increases from 0.1 mg to 1.2 mg.

The N-terminal His-SUMO construct design and the simple, two-step purification procedure depicted in Fig 1D proved effective for all eight ZF arrays tested. To meaningfully compare purity and yield, the cation-exchange step was performed for all eight proteins, regardless of the purity after the affinity step (Fig 1E, lane E). For all proteins, the eluate fractions from cation-exchange chromatography could be pooled and concentrated to 5 mg/mL without precipitation. The total yield ranged from 0.9 to 2.1 mg of protein per liter of culture. SDS-PAGE analysis demonstrated that all products were essentially homogeneous (Fig 3A, top).

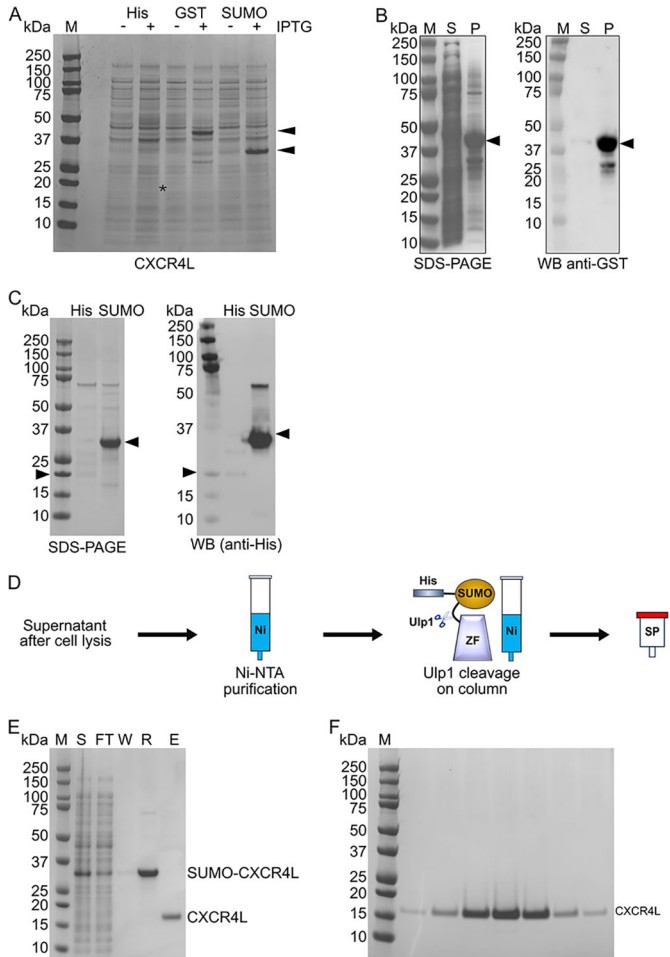

**Fig 1. Expression and purification of ZF arrays. A.** *E. coli* transformed with plasmids encoding His-, GST-, and His-SUMO-tagged CXCR4L (gel lanes labeled "His," "GST," and "SUMO," respectively) before ("–") and after ("+") IPTG induction were loaded on a denaturing polyacrylamide gel and stained with Coomassie Blue. Black asterisk and black arrows indicate the position of the tagged ZF arrays. **B.** Supernatant (S) and pellet (P) fractions of a whole cell lysate containing GST-CXCR4L were separated on a denaturing polyacrylamide gel and stained with Coomassie Blue (left panel) or transferred onto a PVDF membrane and detected with an anti-GST antibody (right panel). **C.** Comparison by SDS-PAGE and western blot (anti-His) of resuspended Ni-NTA resins bound to His-CXCR4L (His, black arrows on the left) or His-SUMO-CXCR4L ("SUMO", black arrows on the right). For each tagged form, the resin volume was kept consistent throughout the procedure. **D.** Schematic representation of the purification protocol. **E.** Samples from each step of the purification protocol were separated on a denaturing polyacrylamide gel and stained with Coomassie Blue. S: Supernatant of cell lysate; FT: Ni-NTA flow-through; W: Ni-NTA washes; R: Ni-NTA resin resuspended in cleavage buffer; E: recovered fraction after overnight cleavage. **F.** Samples spanning the main peak of the cation-exchange chromatographic elution were separated on a denaturing polyacrylamide gel and stained with Coomassie Blue.

To enable fluorescent detection, all constructs included a terminal cysteine residue, which was conjugated to maleimide-activated derivatives of sulfo-cyanine5 ("Cy5") or sulfo-cyanine3 ("Cy3"). Consistent with the poor nucleophilicity of zinc-coordinated thiolates [43–46], labeling occurred exclusively through the terminal cysteine (S1 Fig), resulting in a single product for all ZF arrays (Fig 3A, bottom and 3B). The efficiency of labeling was determined by spectrophotometry to be between 50 and 80% (S2 Table). We next used microscale thermophoresis (MST), a solution-based method that combines infrared heating and fluorescence detection, to confirm sequence-specific DNA recognition by fluorescently labeled ZF arrays. To determine

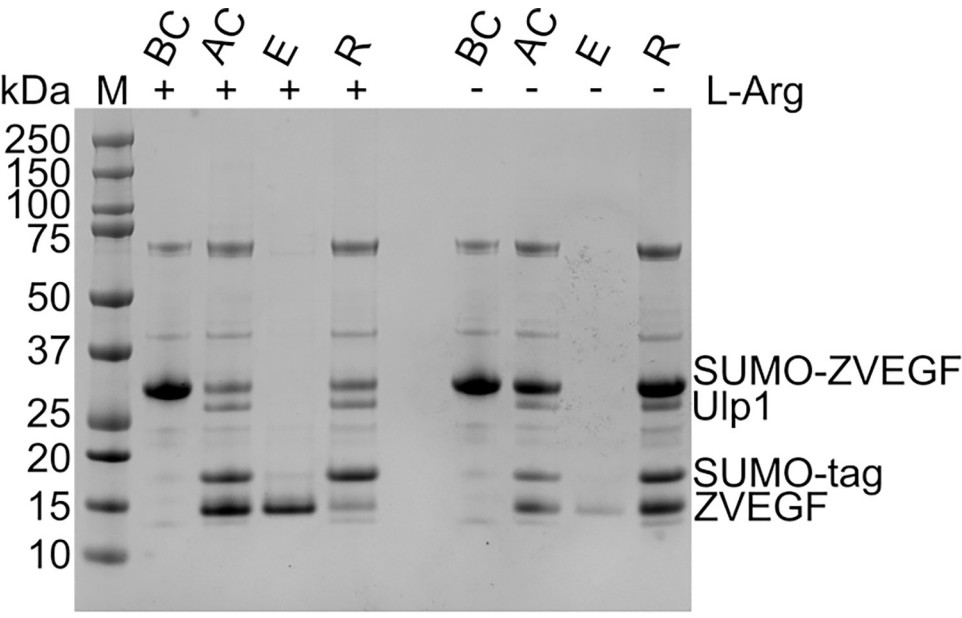

**Fig 2. Effect of L-arginine on Ni-NTA elution.** Analysis by SDS-PAGE of the same steps in presence or not of L-arginine. BC: resin before addition of Ulp1; AC: resin after overnight cleavage; E: recovered fraction after overnight cleavage; R: resin after elution.

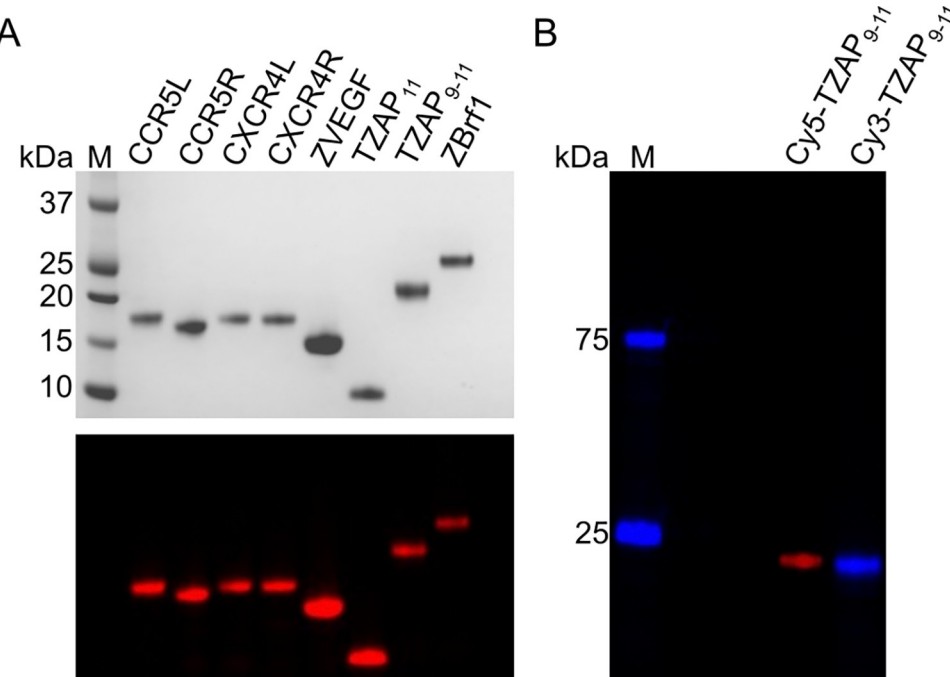

**Fig 3. A general method for the purification and labeling of ZF arrays.** Eight ZF arrays purified by the two-step method outlined in Fig 1 and labeled with maleimide-activated sulfo-Cy5 were separated, alongside molecular weight markers (M), on a denaturing polyacrylamide gel. **A. Top:** the gel after staining with Coomassie Blue. **Bottom:** the same gel upon illumination with 650-nm light. **B.** Cy5- and Cy3-labeled $TZAP_{9-11}$ were electrophoresed through a denaturing polyacrylamide gel and imaged under 650-nm and 532-nm illumination.

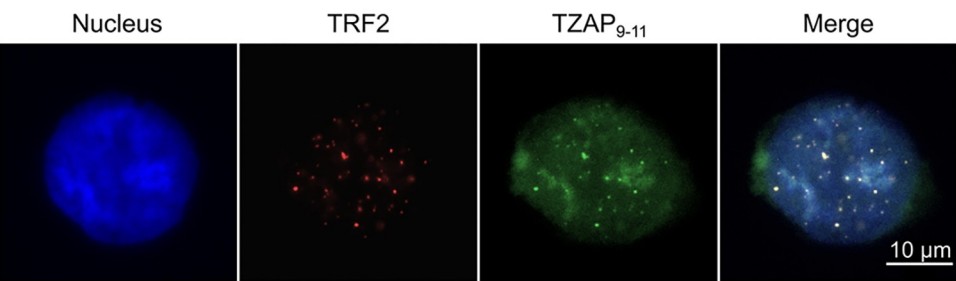

**Fig 4. TZAP$_{9-11}$ binds telomeres *in situ*.** Fixed U2OS cells stained with DAPI (left, blue in the merge image) and incubated with Alexa-Fluor-647-labeled anti-TRF2 (red in the merge image) and Cy3-TZAP$_{9-11}$ (green in the merge image) were imaged on a widefield microscope. Colocalization (yellow) of Cy3-TZAP$_{9-11}$ and anti-TRF2 confirms telomeric localization of TZAP$_{9-11}$. Scale bar: 10 μm.

both the affinity and specificity of DNA binding, a pair of DNA oligonucleotides was designed for each protein, with one oligonucleotide matching the targeted (wild-type, wt) sequence, and the other containing a single point mutation with respect to the wt sequence (S2 Fig).

The measured dissociation constants (K$_D$) ranged from $10^{-7}$ M to $10^{-10}$ M. Importantly, in all cases except for ZBrf1, the affinity for the wt sequence was at least an order of magnitude greater than the affinity for the point mutant (Fig 3), and were consistent with values reported previously using different techniques (TZAP$_{11}$ and TZAP$_{9-11}$, 1.6 x $10^{-7}$ M and 1.8 x $10^{-7}$ nM respectively [47]).

Finally, after confirming sequence-specific binding to naked DNA, we proceeded to test the suitability of labeled ZF arrays as probes for *in situ* detection and localization of DNA. To take advantage of the signal amplification inherent to probes that bind tandem-repeat sequences, we selected for this test a telomere-specific probe, TZAP$_{9-11}$ [47]. U2OS cells, which have relatively long telomeres [48] and are known to bind TZAP [41], were used as a model system. Fixed cells were incubated overnight with the Cy5-TZAP$_{9-11}$ probe at a concentration of 20 nM.

Bright, individual spots were observed in the nucleus (S3A Fig, top panel) and confirmed to be telomeric by their colocalization with TRF2, a telomere-resident protein [49] (Fig 4). No spots were produced in control experiments involving Cy5-maleimide alone (S3A Fig, middle row) or Cy5-ZVEGF, which binds a non-repetitive DNA sequence (S3A Fig, bottom row).

## Discussion

Protocols have been developed for the expression and purification of specific ZF proteins, but none are sufficiently general for direct application to a wide range of ZF arrays; instead, some amount of optimization is typically needed to identify suitable conditions for each new target. For example, although a GST tag results in abundant soluble material for the tetradactyl C2H2 domain of ZNF191 [35], it leads to abundant *insoluble* material for other ZF proteins (Fig 1B). Even though refolding from inclusion bodies is a possibility, it is laborious and has low throughput, and its success is frequently protein dependent [36]. Likewise, the His tag is inconsistently effective for the purification of ZF arrays: it gives satisfactory results for some proteins [25, 27, 28], whereas for others, its use entails problems such as low yield (Fig 1A) or altered conformation [27]. Such difficulties highlighted the need for a robust method which would generalize to new ZF array targets.

After exploring a range of construct designs and purification protocols, we finally arrived at a simple method, involving an N-terminal His-SUMO tag and two chromatographic steps, that consistently yields soluble, functional, and homogeneously pure ZF arrays in milligram

quantities. Indeed, this method proved universally effective for the eight ZF arrays on which it was tested (Fig 2A top). The wide variation in physicochemical properties (pI, MW, hydrophobicity) of the tested C2H2 ZF proteins demonstrates the method's potential to generalize to a broad range of targets.

For nearly two decades the SUMO tag has been used for the expression of challenging proteins [42], but to the best of our knowledge, its application to ZF arrays has not been previously reported. The SUMO tag offers a number of advantages, including increased expression, solubilization, reliable and efficient cleavage, and the potential to recover a native N terminus [42, 50]. For affinity purification, a His tag was fused to the N terminus of SUMO.The addition of L-arginine to storage buffers is known to protect zinc-finger-containing proteins from degradation [51], and protocols for the preparation of zinc-finger nucleases routinely include it following purification [52]. We found that L-arginine confers additional benefits if included throughout the purification process, including two-fold enhanced cleavage efficiency (compare the band for SUMO-ZVGEF in lanes "BC" and "AC" in the presence [+] and absence [−] of L-arginine in Fig 2) and over three-fold decreased retention on Ni-NTA (compare the band for ZVEGF in lanes "AC" and "E" in the presence [+] and absence [−] of L-arginine in Fig 2). These effects are probably attributable, at least in part, to arginine's capacity to prevent protein aggregation and precipitation [53, 54] Retention on Ni-NTA may also be partly due to $Ni^{2+}$ binding by ZFs: although their affinity for nickel is 3–5 logs lower than for zinc [55], the concentration of (chelated) nickel ions in NTA resin is about 200-fold higher than zinc in our buffers (Qiagen LLC).

To adapt these proteins for use as probes, we incorporated a terminal cysteine for facile conjugation to maleimide-activated fluorophores. This approach ensures a compact probe, in contrast to fusion with a fluorescent protein or "self-labeling" enzyme [56, 57], whose bulk could hinder inter- or intranucleosomal binding. Fluorescent labeling enabled straightforward thermophoretic measurement of DNA affinity. Labeled ZF arrays exhibited high affinity for DNA (S4 Table), comparable to literature-reported values, where available [47]. Seven of the eight ZF arrays bound their target DNA specifically, with affinity falling by at least an order of magnitude upon mutation of a single base pair in the target sequence (Fig 5 and S4 Table). In contrast, ZBrf1 has less stringent DNA preferences, binding equally well to point mutants as to its intended target sequence (S4 Table). It is nonetheless effective for *in situ* labeling [17] as the accumulation of fluorescent signal at the tandemly repeated locus compensates for limited specificity.

Cy3-TZAP$_{9-11}$ can be used to label telomeres in fixed cells. However, a high background of Cy3-TZAP$_{9-11}$ was observed, which was not present in control experiments involving Cy3 dye alone (S3B Fig). This background can be explained by specific binding to extra-telomeric TTAGGG sequences, which will occur by chance approximately every 4 kilobases, and by nonspecific binding to related sequences. A similar level background was produced by other ZF arrays (S3A Fig, bottom row).

Having established a fast and reliable procedure for the purification of arbitrary ZF arrays, it becomes feasible to prepare ZF array libraries, which would in turn enable the detection of nonrepetitive sequences in chromosomal DNA. Tandemly repeated DNA, which accounts for 7 to 8% of the human genome [58], can be readily detected by *in situ* labeling approaches, as tandem repetition provides a convenient mechanism for signal amplification. The overwhelming majority of the human genome, however, is not tandemly repeated and remains difficult to detect, because of either insufficient analytical sensitivity or insufficient probe specificity. In either case, the limitation can be addressed by the use of a probe library whose members bind adjacent sequences. Such an approach has been demonstrated using oligonucleotide hybridization [59] and CRISPR/dCas9 [60], but the former requires denaturation and the latter entails

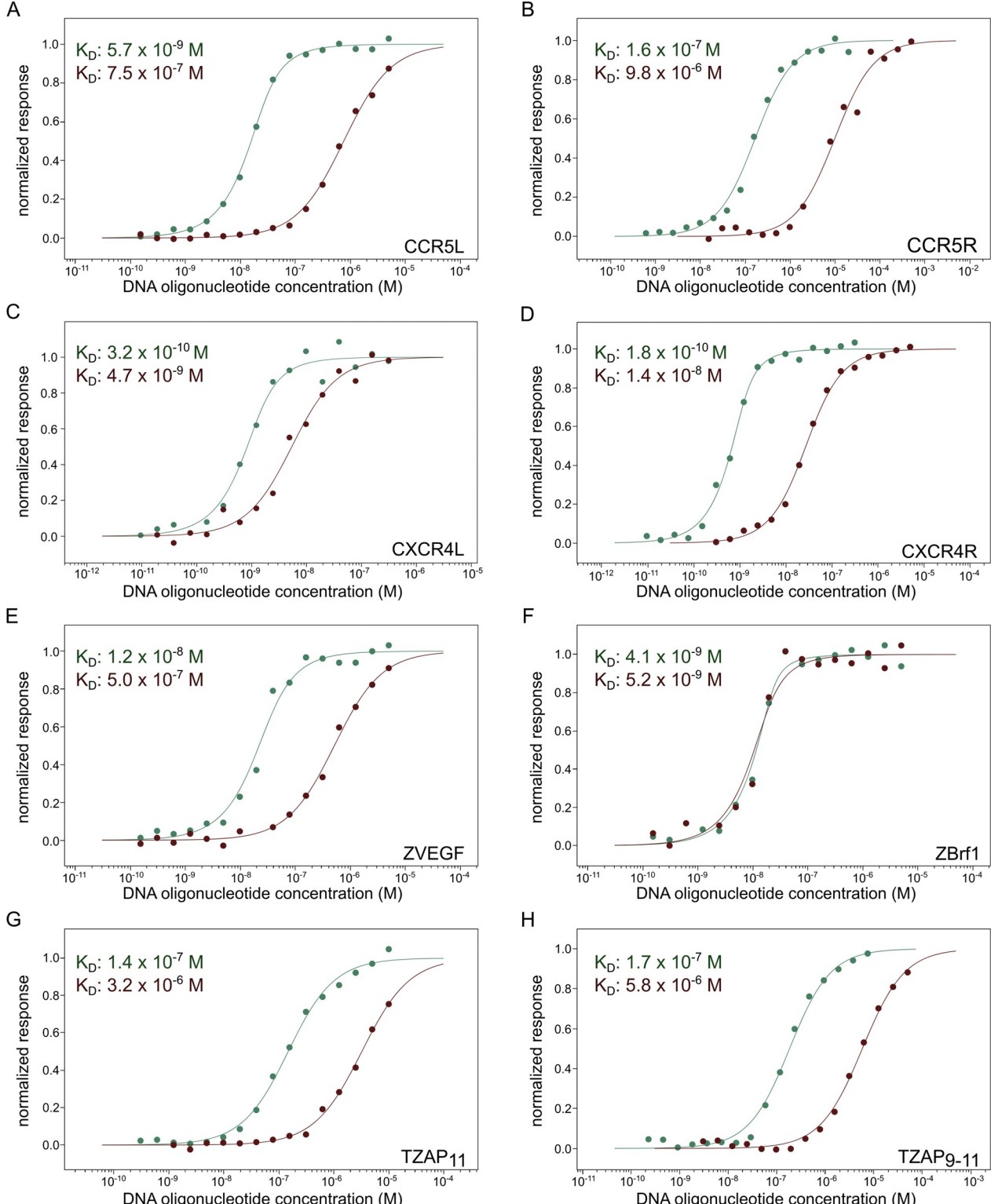

**Fig 5. Affinity and specificity of ZF arrays for DNA, measured by MST.** Each panel shows a binding curve involving the ZF array indicated in the bottom right corner and wild-type (green) or point-mutant (red) DNA sequences. The ordinate represents fractional saturation and the abscissa represents the concentration of the oligonucleotide. Solid lines represent fits to a non-cooperative 1:1 binding model.

DNA melting which is likely to perturb the surrounding chromatin structure through torsional effects. Furthermore, CRISPR-based labeling is bound to have major steric consequences, given the size of dCas9 (~160 kDa) and a single-guide RNA (~30 kDa), which are together comparable to a nucleosome (~200 kDa). In contrast, zinc fingers are among the most compact DNA-binding proteins, requiring only 1 kDa per base pair recognized (compared to ~10 kDa per base pair for CRISPR-based approaches). ZF proteins can even bind intranucleosomal sites [61] that are inaccessible to Cas9 [62]. ZF proteins therefore provide a viable route to determining the native structure of defined chromosomal regions, and the method we describe makes this route tractable.

## Material and methods

### ZF array expression

The DNA sequences for the eight ZF arrays (Table 1 and S1 Table) were codon-optimized for *E. coli* expression and cloned into pET28b (+) for His-tagged proteins, pGEX2T for GST-tagged proteins, or pET28-SUMO for His-SUMO-tagged proteins. The resulting plasmids were transformed into *E. coli* BL21 (DE3). The cells were grown in LB medium—supplemented with 100 μM $ZnCl_2$ and appropriate antibiotics (50 μg/mL kanamycin for His- and His-SUMO-tagged proteins or 100 μg/mL ampicillin for GST-tagged proteins)—at 37˚C and 180 rpm until the optical density at 600 nm (OD600) reached 0.7. Protein expression was induced by adding 0.3 mM IPTG and incubating at 16˚C overnight. After induction, the cells were harvested and resuspended in 10 mL of lysis buffer (20 mM Tris-HCl [pH 7.5], 2 M NaCl, 100 μM $ZnCl_2$, 10 mM imidazole, supplemented with PMSF and protease inhibitors before lysis) per gram of wet cell mass [24] Cells were then lysed by sonication (50% power, 15 s on, 45 s off, 5 min), and the lysate was clarified by centrifugation at 60,000 g for 1 h at 4˚C.

### Purification of His- and His-SUMO-tagged ZF arrays

The clarified lysate was loaded onto Ni-NTA resin (1 mL bed volume per liter of cell culture) which had been pre-equilibrated with lysis buffer. The resin was washed with ten bed volumes of wash buffer (20 mM Tris-HCl pH 7.5, 500 mM NaCl, 100 μM $ZnCl_2$, 60 mM imidazole). For SDS-PAGE and western blot analysis, the resin containing His-tagged ZF proteins was resuspended in five bed volumes of cleavage buffer (20 mM Tris-HCl pH 7.5 or pH 7.0 [for $TZAP_{11}$ and $TZAP_{9-11}$], 100 mM NaCl, 100 μM $ZnCl_2$, 5 mM β-mercaptoethanol [β-ME], 100 mM L-arginine). Ni-NTA resin containing His-SUMO-ZF proteins was resuspended in five bed volumes of cleavage buffer. Ulp1 protease was added to the resin at a ratio of 0.1 mg of protease per 1 mL of resin, and the mixture was rotated at 4˚C overnight to cleave the His-SUMO tag.

Following cleavage, the flow-through containing the protein was loaded onto a 5-mL HiTrap SP HP column, which had been pre-equilibrated with two column volumes of cleavage buffer. The protein was eluted with a linear gradient of sodium chloride from 100 mM to 1 M. Protein purity was assessed by SDS-PAGE (Invitrogen NuPAGE™ 4–12% Bis-Tris, MES running buffer) with Coomassie staining. The purified proteins were concentrated to 1 mg/mL, flash frozen in liquid nitrogen, and stored at −80˚C.

### ZF array conjugation

$TZAP_{9-11}$ already contains two cysteine residues at its C-terminus. For the other ZF arrays, a cysteine was added to either the amino terminus (ZVEGF and $TZAP_{11}$) or the carboxy terminus (CCR5L, CCR5R, CXCR4L, CXCR4R, and ZBrf1). Maleimide-activated sulfo-cyanine 5

("Cy5") or sulfo-cyanine 3 ("Cy3") (both from Lumiprobe) were dissolved in DMSO to a concentration of 10 mM. ZF arrays were diluted to 0.5 mg/mL in degassed labeling buffer (same as cleavage buffer but without β-ME) supplemented with 0.1 mM TCEP (in lieu of β-ME). A five-fold molar excess of dye was added dropwise to the protein solution in the dark while stirring. The reaction was allowed to proceed on a rotator at room temperature for two hours. To quench the maleimide group of unreacted dye, a ten-fold molar excess of β-ME was added.

ZF arrays and free dye were separated on a 1-mL HiTrap SP HP column (Cytiva) that had been pre-equilibrated with five column volumes of labeling buffer. ZF arrays were eluted using a linear gradient of sodium chloride from 100 mM to 1 M. The concentrations of ZVEGF, $TZAP_{11}$, $TZAP_{9-11}$, and ZBrf1 were determined by measuring absorbance at 280 nm (A280), with correction for the contribution of dye at this wavelength. The concentrations of the dyes were determined by absorption measurements at their respective excitation wavelengths: for Cy3, 548 nm ($\varepsilon$ = 162000 L • mol-1 • cm-1); for Cy5, 662 nm ($\varepsilon$ = 271000 L • mol-1 • cm-1). For CCR5L, CCR5R, CXCR4L, and CXCR4R, whose molar extinction coefficient is effectively zero (in the reduced state), protein concentration was determined according to the method of Bradford. The labeling efficiency was calculated as the ratio of the molar concentration of the dye to the molar concentration of protein. Labeling controls consisted of cystein-free terminus ZF arrays.

### Microscale Thermophoresis (MST)

With the exception of the tzap-wt and tzap-mut sequences, for which double-stranded DNA sequences [47] were used, wild-type DNA sequences, and variants bearing single point mutations, were incorporated into hairpin structures (S1 Fig).

DNA oligonucleotides were dissolved to a concentration of 500 μM in MST binding buffer (20 mM Tris-HCl [pH 7.5], 200 mM NaCl, 0.01% v/v Tween-20, 100 μM $ZnCl_2$, 0.1 mM TCEP). Annealing was performed by heating at 95˚C for 2 min and then allowing samples to slowly cool to room temperature. Cy5-conjugated ZF arrays were diluted to the concentrations listed in S3 Table; for $TZAP_{11}$ and $TZAP_{9-11}$, the diluent was TZAP buffer (20 mM Tris-HCl [pH 7.0], 50 mM NaCl, 0.1% v/v Tween-20, 100 μM $ZnCl_2$, 0.1 mM TCEP), and for the remaining ZF arrays, the diluent was MST binding buffer. A two-fold dilution series with 16 concentrations was prepared for each DNA oligonucleotide using TZAP buffer (for tzap-wt and tzap-mut) or MST binding buffer (for hairpins). The concentration ranges examined are specified in S3 Table. The ZF arrays and serially diluted oligonucleotides were mixed and loaded into a capillary for measurement on a Monolith Pico system (NanoTemper Technologies). The powers of the LED and IR laser were optimized automatically by the system software. The resulting data were analyzed with the Monolith NT Analysis Software.

### Fixed cell labeling

Fluorescent labeling of fixed cells was performed as previously described [63]. Briefly, U2OS cells were seeded onto 12-mm poly-lysine-coated coverslips (Neuvitro Corporation, cat: GG-12-1.5-PDL) at a density of 2 x $10^5$ cells/mL and cultured in McCoy's 5A medium at 37˚C with 5% $CO_2$ overnight. Cells were fixed at −20˚C for 20 min using a pre-chilled solution of 1:1 (v/v) methanol and acetic acid. The fixed samples were washed three times with PBS (5 min per wash). The coverslips were incubated in blocking buffer (20 mM Tris-HCl [pH 7.5], 150 mM NaCl, 0.1% v/v Tween-20, 100 μM $ZnCl_2$, 0.1 mM TCEP, 1% w/v BSA) for 1 h at 37˚C. The coverslips were incubated overnight at 4˚C in blocking buffer supplemented with 20 nM Cy3- or Cy5-labeled ZF arrays (or the dyes alone quenched with 10 excess fold of β-ME). After incubation, the coverslips were washed three times with PBS (5 min per wash) and mounted with

VECTASHIELD Antifade Mounting Medium containing DAPI (Vector Laboratories). Colocalization experiments were performed similarly, using an anti-TRF2 antibody labeled with Alexa Fluor 647 (Novus Biologicals) diluted 1:100 in blocking buffer, and incubated with the fixed cells overnight at 4°C, along with 20 nM Cy3-conjugated TZAP$_{9-11}$.

## Microscopy and image analysis

Fixed cells were imaged on an inverted microscope (Nikon Microscope Eclipse 80i) equipped with a Nikon Plan Apo 60xA oil IR objective and a CCD camera (Spot Insight 4.0 Mp Model 16.0). Fluorescent signal was recorded using white-light excitation in combination with the appropriate filter cube set for DAPI, Cy3, or Cy5 (Alexa Fluor 647). Image analysis was performed using ImageJ software.

## Immunoblotting

The 60,000 g pellet after cell lysis was resuspended in lysis buffer to the same volume as the starting lysate. Two microliters of the lysate supernatant and resuspended pellet were mixed with loading buffer and boiled for 5 min. Samples were run on NuPAGE™ 4–12% Bis-Tris gels (Invitrogen) and transferred to PVDF membrane using the Trans-Blot Turbo Transfer system (Bio-Rad). Membranes were blocked in Tris-buffered saline solution (25 mM Tris-HCl, [pH 7.5], 2.7 mM KCl, 137 mM NaCl) supplemented with 0.1% v/v Tween-20 (TBST) and 5% w/v non-fat dry milk for 1 h at room temperature. The membrane was washed thrice with TBST and incubated at room temperature for 1 h with HRP-conjugated anti-GST (Abcam, ab3416) or HRP-conjugated anti-His antibody (Proteintech, HRP-66005) at 1:10,000 dilution. After washing thrice with TBST, the membrane was developed using the enhanced chemiluminescence technique described in the manual of the Immobilon ECL UltraPlus Western HRP Substrate (Millipore).

## Supporting information

**S1 Fig. ZF arrays without free cysteine cannot be labeled with maleimide-activated dye.**
ZVEGF with (+) and without (-) free cysteine were labeled with Cy5and compared by SDS-PAGE.
(TIFF)

**S2 Fig. Hairpin and double-stranded DNA constructs used for MST.** Black boxes indicate the binding site for ZF proteins, and black arrows indicate mutated sites.
(TIFF)

**S3 Fig. Labeling of telomeres *in situ* with TZAP$_{9-11}$ compared to non-specific binding. A. Top**: Fixed U2OS cells stained with DAPI (blue) and labeled with Cy5-TZAP$_{9-11}$ probe (red) were imaged on a widefield microscope. **Middle**: Fixed U2OS cells stained with DAPI (blue) and labeled with Cy5 (quenched with β-ME, red) were imaged on a widefield microscope. **Bottom**: Fixed U2OS cells stained with DAPI (blue) and labeled with Cy5-ZVEGF (red) were imaged on a widefield microscope. **B**. Fixed U2OS cells stained with DAPI (blue) and labeled with Cy3 (quenched with β-ME) were imaged on a widefield microscope. Scale bar: 10 μm.
(TIFF)

**S4 Fig. Raw images.**
(PDF)

**S1 Table. Sequences of ZF arrays.**
(DOCX)

**S2 Table. Efficiencies of fluorophore coupling to ZF arrays.**
(DOCX)

**S3 Table. Concentrations of fluorescent ZF arrays and DNA oligonucleotides used for MST measurements.**
(DOCX)

**S4 Table. DNA binding results of ZF arrays.**
(DOCX)

## Acknowledgments

We thank Dr. Jia Liu for providing initial plasmids for His-ZFPs. We thank Dr. Heqiao Zhang for providing suggestions. We thank Ms. Xiuxia Gao and Ms. Feifan Wen for their contributions during the initial stage of this research.

## Author Contributions

**Conceptualization:** Roger D. Kornberg, Andrew J. Beel, Pierre-Jean Mattei.

**Investigation:** Jingchang Liang, Guanqiao Wang.

**Methodology:** Jingchang Liang, Maia Azubel, Roger D. Kornberg, Andrew J. Beel, Pierre-Jean Mattei.

**Resources:** Yan Nie, Roger D. Kornberg.

**Supervision:** Maia Azubel, Yan Nie, Roger D. Kornberg, Andrew J. Beel, Pierre-Jean Mattei.

**Writing – original draft:** Jingchang Liang, Pierre-Jean Mattei.

**Writing – review & editing:** Jingchang Liang, Maia Azubel, Yan Nie, Roger D. Kornberg, Andrew J. Beel, Pierre-Jean Mattei.

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
