## [Decision Letter · Decision Letter 0]

20 Nov 2024

PONE-D-24-50585A universal method for zinc finger protein purificationPLOS ONE

Dear Dr. Mattei,

Thank you for submitting your manuscript to PLOS ONE. After careful consideration, we feel that it has merit but does not fully meet PLOS ONE’s publication criteria as it currently stands. Therefore, we invite you to submit a revised version of the manuscript that addresses the points raised during the review process.

We look forward to receiving your revised manuscript.

Kind regards,

Yu-Hsuan Tsai

Academic Editor

PLOS ONE

Journal Requirements:

3. Thank you for stating the following financial disclosure: [AJB was funded by an NIH grant 1DP5OD033431 J.L was funded by WLA laboratories. Y.N. and G.W. were funded by a grant from ShanghaiTech University.]. Please state what role the funders took in the study. If the funders had no role, please state: "The funders had no role in study design, data collection and analysis, decision to publish, or preparation of the manuscript." If this statement is not correct you must amend it as needed. Please include this amended Role of Funder statement in your cover letter; we will change the online submission form on your behalf.

4. Thank you for stating the following in your Competing Interests section: [copy in statement]. Please complete your Competing Interests on the online submission form to state any Competing Interests. If you have no competing interests, please state "The authors have declared that no competing interests exist.", as detailed online in our guide for authors at http://journals.plos.org/plosone/s/submit-now This information should be included in your cover letter; we will change the online submission form on your behalf.

5. We note that your Data Availability Statement is currently as follows: [All relevant data are within the manuscript and its Supporting Information files] Please confirm at this time whether or not your submission contains all raw data required to replicate the results of your study. Authors must share the “minimal data set” for their submission. PLOS defines the minimal data set to consist of the data required to replicate all study findings reported in the article, as well as related metadata and methods (https://journals.plos.org/plosone/s/data-availability#loc-minimal-data-set-definition). For example, authors should submit the following data: - The values behind the means, standard deviations and other measures reported; - The values used to build graphs; - The points extracted from images for analysis. Authors do not need to submit their entire data set if only a portion of the data was used in the reported study. If your submission does not contain these data, please either upload them as Supporting Information files or deposit them to a stable, public repository and provide us with the relevant URLs, DOIs, or accession numbers. For a list of recommended repositories, please see https://journals.plos.org/plosone/s/recommended-repositories. If there are ethical or legal restrictions on sharing a de-identified data set, please explain them in detail (e.g., data contain potentially sensitive information, data are owned by a third-party organization, etc.) and who has imposed them (e.g., an ethics committee). Please also provide contact information for a data access committee, ethics committee, or other institutional body to which data requests may be sent. If data are owned by a third party, please indicate how others may request data access.

6. PLOS ONE now requires that authors provide the original uncropped and unadjusted images underlying all blot or gel results reported in a submission’s figures or Supporting Information files. This policy and the journal’s other requirements for blot/gel reporting and figure preparation are described in detail at https://journals.plos.org/plosone/s/figures#loc-blot-and-gel-reporting-requirements and https://journals.plos.org/plosone/s/figures#loc-preparing-figures-from-image-files. When you submit your revised manuscript, please ensure that your figures adhere fully to these guidelines and provide the original underlying images for all blot or gel data reported in your submission. See the following link for instructions on providing the original image data: https://journals.plos.org/plosone/s/figures#loc-original-images-for-blots-and-gels. In your cover letter, please note whether your blot/gel image data are in Supporting Information or posted at a public data repository, provide the repository URL if relevant, and provide specific details as to which raw blot/gel images, if any, are not available. Email us at plosone@plos.org if you have any questions.

Reviewers' comments:

Reviewer's Responses to Questions

**Comments to the Author**

1. Is the manuscript technically sound, and do the data support the conclusions?

Reviewer #1: Yes

Reviewer #2: Yes

2. Has the statistical analysis been performed appropriately and rigorously? 

Reviewer #1: Yes

Reviewer #2: I Don't Know

3. Have the authors made all data underlying the findings in their manuscript fully available?

Reviewer #1: Yes

Reviewer #2: Yes

4. Is the manuscript presented in an intelligible fashion and written in standard English?

Reviewer #1: Yes

Reviewer #2: Yes

5. Review Comments to the Author

Reviewer #1: Liang et.al. reported a universal method for zinc finger protein purification involving two chromatographic steps. As we know, there are different groups of zinc fingers based on the overall shapes pf the protein backbone in the folded domain. Here authors only investigated C2H2 ZF proteins. Does this method apply to other types of protein, such as C4 or C6 ZF proteins? Moreover, purification was the main focus of the studies and the importance of buffer conditions used in purification process was emphasized in the discussion section. However, there is no detailed information in the main text or figures to show this part. Therefore, the manuscript should be revised accordingly before being accepted for publication.

Major points

1. Lack of information related to the optimization of buffer conditions, such as adding L-arginine before or after purification.

2. Lack additional experiments to validate the method to other types of ZF proteins.

Minor points

1. The figures quality can be improved with higher resolution.

2. Line 55, Remove e.g..

3. Line 66-72, “A general method for the purification of ZF…of probes for non-repetitive sequences.” Should move before Line 55.

4. Line 120-121, Wrong place for sentence “IPTG induction led to a prominent band of the expected molecular weight in whole cell lysates (Fig. 1A, compare lanes with and without IPTG).”

5. Fig 1E, the size of the CXCR4L was slightly not consistent in two SDS-PAGE. Why not just simply show the Marker band in the gel too?

6. Line 170, control experiments are needed in Fig.S2. A non-specific probe should be included.

7. Fig.4. is there non-specific binding of TZAP9-11 in telomeres (shown in green)? Whether this signal (extra green signal) comes from Cy3 signal?

8. Line 257, remove second “pH7.5”.

9. Line 260 and line 263, the purpose of using two different buffer, 100 mM L-Arg-HCl vs 100 mM L-Arg.

10. Why only show TZAP9-11 in Fig. 2B?

11. Information related to yield of each 8 tested proteins can be added to table 1.

Reviewer #2: In this manuscript, the authors reported recombinant preparations of various zinc fingers, by use of SUMO fusion. They confirm their production and characterize these ZFs with cysteine labelling and in situ labelling.

Comments:

-The location of cysteine labelling is not confirmed. The authors should have tested the labelling with the WT ZFs. Mass spectrometry analysis should be included.

-The Kd constants should be compared with the literature values, if available.

-In the introduction, the authors should provide a reasoning why ZFs are difficult to produce. Also, an explanation to why SUMO is superior than others should be included.

6. PLOS authors have the option to publish the peer review history of their article (what does this mean?). If published, this will include your full peer review and any attached files.

Reviewer #1: No

Reviewer #2: No

---

## [Author Response · Author response to Decision Letter 0]

2 Jan 2025

Dear editors and reviewers, 

Thank you very much for your time and your constructive feedback on our manuscript; it has been revised and improved accordingly. Please find below the reviewers’ and academic editor’s comments (in black font) interleaved with our responses (in light blue font).

Best,

Andrew J. Beel and Pierre-Jean Mattei

The manuscript is in accord with PLOS ONE’s style requirements.

The Funding Information section has been updated.

3. Thank you for stating the following financial disclosure: [AJB was funded by an NIH grant 1DP5OD033431 J.L was funded by WLA laboratories. Y.N. and G.W. were funded by a grant from ShanghaiTech University.]. Please state what role the funders took in the study. If the funders had no role, please state: "The funders had no role in study design, data collection and analysis, decision to publish, or preparation of the manuscript." If this statement is not correct you must amend it as needed. Please include this amended Role of Funder statement in your cover letter; we will change the online submission form on your behalf.

The role of the funding sources has been added.

4. Thank you for stating the following in your Competing Interests section: [copy in statement]. Please complete your Competing Interests on the online submission form to state any Competing Interests. If you have no competing interests, please state "The authors have declared that no competing interests exist.", as detailed online in our guide for authors at http://journals.plos.org/plosone/s/submit-now This information should be included in your cover letter; we will change the online submission form on your behalf.

The Competing Interests section has been updated in the paper and the cover letter.

5. We note that your Data Availability Statement is currently as follows: [All relevant data are within the manuscript and its Supporting Information files] Please confirm at this time whether or not your submission contains all raw data required to replicate the results of your study. Authors must share the “minimal data set” for their submission. PLOS defines the minimal data set to consist of the data required to replicate all study findings reported in the article, as well as related metadata and methods (https://journals.plos.org/plosone/s/data-availability#loc-minimal-data-set-definition). For example, authors should submit the following data: - The values behind the means, standard deviations and other measures reported; - The values used to build graphs; - The points extracted from images for analysis. Authors do not need to submit their entire data set if only a portion of the data was used in the reported study. If your submission does not contain these data, please either upload them as Supporting Information files or deposit them to a stable, public repository and provide us with the relevant URLs, DOIs, or accession numbers. For a list of recommended repositories, please see https://journals.plos.org/plosone/s/recommended-repositories. If there are ethical or legal restrictions on sharing a de-identified data set, please explain them in detail (e.g., data contain potentially sensitive information, data are owned by a third-party organization, etc.) and who has imposed them (e.g., an ethics committee). Please also provide contact information for a data access committee, ethics committee, or other institutional body to which data requests may be sent. If data are owned by a third party, please indicate how others may request data access.

Raw data are included in the supplement.

6. PLOS ONE now requires that authors provide the original uncropped and unadjusted images underlying all blot or gel results reported in a submission’s figures or Supporting Information files. This policy and the journal’s other requirements for blot/gel reporting and figure preparation are described in detail at https://journals.plos.org/plosone/s/figures#loc-blot-and-gel-reporting-requirements and https://journals.plos.org/plosone/s/figures#loc-preparing-figures-from-image-files. When you submit your revised manuscript, please ensure that your figures adhere fully to these guidelines and provide the original underlying images for all blot or gel data reported in your submission. See the following link for instructions on providing the original image data: https://journals.plos.org/plosone/s/figures#loc-original-images-for-blots-and-gels. In your cover letter, please note whether your blot/gel image data are in Supporting Information or posted at a public data repository, provide the repository URL if relevant, and provide specific details as to which raw blot/gel images, if any, are not available. Email us at plosone@plos.org if you have any questions.

Uncropped data are present in the supporting information.

The titles and captions for the Supporting Information have been added to the manuscript.

5. Review Comments to the Author

Reviewer #1: Liang et.al. reported a universal method for zinc finger protein purification involving two chromatographic steps. As we know, there are different groups of zinc fingers based on the overall shapes pf the protein backbone in the folded domain. Here authors only investigated C2H2 ZF proteins. Does this method apply to other types of protein, such as C4 or C6 ZF proteins? 

Moreover, purification was the main focus of the studies and the importance of buffer conditions used in purification process was emphasized in the discussion section. However, there is no detailed information in the main text or figures to show this part. Therefore, the manuscript should be revised accordingly before being accepted for publication.

Major points

1. Lack of information related to the optimization of buffer conditions, such as adding L-arginine before or after purification.

We have added a gel which shows that the addition of L-arginine to the cleavage buffer has two important effects (Fig 2). It increases the efficiency of digestion (compare the band intensities in lanes “BC” and “AC”), and it prevents nonspecific binding of cleaved ZF protein to the resin (compare lanes “AC” and “E”, as well as lane “R”), allowing for significantly increased recovery (compare lanes “E”). Prior studies on zinc-finger nucleases (e.g., Liu et al., Mol. Ther. Nucleic Acids, 2015) have added arginine after purification to prevent degradation (cf. Zhao & Huang, Chin. Chem. Lett., 2007). 

2. Lack additional experiments to validate the method to other types of ZF proteins.

We have not tested this method on other types of ZF proteins. C2H2 ZF proteins are by far the most abundant ZF proteins and the best studied. We have added “C2H2” to the title to better reflect our aim.

Minor points

1. The figures quality can be improved with higher resolution.

Higher resolution images have been uploaded with the revised manuscript.

2. Line 55, Remove e.g..

It has been removed.

3. Line 66-72, “A general method for the purification of ZF…of probes for non-repetitive sequences.” Should move before Line 55.

While we appreciate the reviewer's suggestion to adjust the placement of this note, we believe the current structure better supports the logical progression of our argument. The note about nonrepetitive loci serves to introduce an important scenario that current purification methods cannot effectively address. Existing methods often require extensive optimization for each individual ZF protein. This challenge is magnified for nonrepetitive loci, where designing multiple ZF proteins necessitates independent optimization for each, a process which is laborious and impractical. By placing this note in the context of the limitations of current purification methods, we emphasize the need for a more general approach.

The current structure begins by introducing the imperfection of ZF modularity and how recent advances in machine learning have facilitated the design of polydactyl ZF proteins. This naturally leads to the practical bottleneck: the lack of generality of existing purification methods. Placing the note about nonrepetitive loci here ties it directly to this bottleneck, illustrating how current limitations hinder specific applications.

If we were to move this note earlier, before discussing the limitations of current methods, we feel its significance would be less apparent, lacking the context of case-by-case optimization which is often necessary. By introducing it where we do, we maintain a logical progression: outlining the technical advances, addressing the practical bottlenecks, and illustrating their significance with a concrete example. This structure allows for a seamless transition into the Results section. We hope this explanation clarifies the reasoning behind our placement of this paragraph.

4. Line 120-121, Wrong place for sentence “IPTG induction led to a prominent band of the expected molecular weight in whole cell lysates (Fig. 1A, compare lanes with and without IPTG).”

This sentence has been relocated.

5. Fig 1E, the size of the CXCR4L was slightly not consistent in two SDS-PAGE. Why not just simply show the Marker band in the gel too?

The MW markers are now shown in the figure.

6. Line 170, control experiments are needed in Fig.S2. A non-specific probe should be included.

We have included the results of control experiments in a supplementary figure (S4 Fig); these show the fluorescence produced by dye alone and by Cy5-ZVEGF, a ZF protein that binds to a non-repeated DNA sequence.

7. Fig.4. is there non-specific binding of TZAP9-11 in telomeres (shown in green)? Whether this signal (extra green signal) comes from Cy3 signal?

Yes, there is also off-target binding; this is to be expected, as the cognate sequence can occur extra-telomerically, and probes may bind to related sequences (point mutations generally decrease affinity by about an order of magnitude). The telomeres are visible against the background because the telomeric repeats locally concentrate probes. Control experiments have also been added (see S4 Fig) showing that Cy3 alone produces no signal.

8. Line 257, remove second “pH7.5”.

This typographical error has been corrected.

9. Line 260 and line 263, the purpose of using two different buffer, 100 mM L-Arg-HCl vs 100 mM L-Arg.

This typographical error has been corrected.

10. Why only show TZAP9-11 in Fig. 2B?

ZF arrays other than TZAP9-11 bind unique sequences and would not be detectable by light microscopy. See also the response to point 7.

11. Information related to yield of each 8 tested proteins can be added to table 1.

Yield information has been added to Table 1.

Reviewer #2: In this manuscript, the authors reported recombinant preparations of various zinc fingers, by use of SUMO fusion. They confirm their production and characterize these ZFs with cysteine labelling and in situ labelling.

Comments:

-The location of cysteine labelling is not confirmed. The authors should have tested the labelling with the WT ZFs. Mass spectrometry analysis should be included.

The site of labeling was confirmed by experiments involving ZF proteins that lack a free cysteine residue: these samples are not labeled by maleimide-activated dyes (S2 Fig).

-The Kd constants should be compared with the literature values, if available.

Only for TZAP11 and TZAP9-11 are dissociation constants available for comparison (these values were added to the main text).

-In the introduction, the authors should provide a reasoning why ZFs are difficult to produce. Also, an explanation to why SUMO is superior than others should be included.

The reviewer raises an interesting question about the reason for the lack of generality of published protocols. Unfortunately, providing an answer is not straightforward, for even small changes to constructs can sometimes have large effects on expression levels or solubility. As a result, it is standard practice in biochemistry to "optimize" constructs and conditions for high yield of soluble protein. This typically involves exploration of different tags, linkers, tag placement, buffers, and so forth. Recognizing that this practice would be a hindrance to the use of designed ZF arrays for genome-targeting applications, we sought to identify a protocol which would work well for a variety of ZF proteins. One could speculate on the reasons for the success of our method (e.g., the SUMO tag stabilizes ZF proteins in bacterial cells), but actually determining the reasons would entail a study unto itself.

6. PLOS authors have the option to publish the peer review history of their article (what does this mean?). If published, this will include your full peer review and any attached files.

That is fine.

---

## [Decision Letter · Decision Letter 1]

14 Jan 2025

A universal method for the purification of C2H2 zinc finger arrays

PONE-D-24-50585R1

Dear Dr. Mattei,

We’re pleased to inform you that your manuscript has been judged scientifically suitable for publication and will be formally accepted for publication once it meets all outstanding technical requirements.

Kind regards,

Yu-Hsuan Tsai

Academic Editor

PLOS ONE

Additional Editor Comments (optional):

Reviewers' comments:

Reviewer's Responses to Questions

**Comments to the Author**

1. If the authors have adequately addressed your comments raised in a previous round of review and you feel that this manuscript is now acceptable for publication, you may indicate that here to bypass the “Comments to the Author” section, enter your conflict of interest statement in the “Confidential to Editor” section, and submit your "Accept" recommendation.

Reviewer #1: All comments have been addressed

Reviewer #2: All comments have been addressed

2. Is the manuscript technically sound, and do the data support the conclusions?

Reviewer #1: Yes

Reviewer #2: (No Response)

3. Has the statistical analysis been performed appropriately and rigorously? 

Reviewer #1: Yes

Reviewer #2: Yes

4. Have the authors made all data underlying the findings in their manuscript fully available?

Reviewer #1: Yes

Reviewer #2: Yes

5. Is the manuscript presented in an intelligible fashion and written in standard English?

Reviewer #1: Yes

Reviewer #2: Yes

6. Review Comments to the Author

Reviewer #1: The authors have addressed all the questions I have asked in previous revision process. The current version of manuscript is satisfied for the publication.

Reviewer #2: My concerns have been addressed on cysteine labelling, Kd values, and difficulties on ZFs production. The article is suitable for publication.

7. PLOS authors have the option to publish the peer review history of their article (what does this mean?). If published, this will include your full peer review and any attached files.

Reviewer #1: No

Reviewer #2: No

---

## [Editor Report · Acceptance letter]

24 Jan 2025

PONE-D-24-50585R1 

PLOS ONE

Dear Dr. Mattei, 

I'm pleased to inform you that your manuscript has been deemed suitable for publication in PLOS ONE. Congratulations! Your manuscript is now being handed over to our production team.

Kind regards, 

on behalf of

Dr. Yu-Hsuan Tsai 

Academic Editor

PLOS ONE